# Efficacy and Safety of Useul for Dry Eye Disease: Protocol for a Randomized, Double-Blind, Placebo-Controlled, Parallel, Phase 2 Clinical Trial

**DOI:** 10.3390/healthcare12232383

**Published:** 2024-11-27

**Authors:** Yee-Ran Lyu, O-Jin Kwon, Bongkyun Park, Hyun-A Jung, Ga-Young Lee, Chan-Sik Kim

**Affiliations:** 1Korean Medicine Science Research Division, Korea Institute of Oriental Medicine, Daejeon 34054, Republic of Korea; onedoctor2ran@kiom.re.kr (Y.-R.L.); cheda1334@kiom.re.kr (O.-J.K.); 2Korean Medicine Convergence Research Division, Korea Institute of Oriental Medicine, Daejeon 34054, Republic of Korea; bkpark@kiom.re.kr; 3Department of Oriental Ophthalmology, Otolaryngology & Dermatology, College of Korean Medicine, Daejeon University, Daejeon 35235, Republic of Korea; acua3739@dju.kr; 4BTGIN Company, Daejeon 35235, Republic of Korea; rnd1@btgin.com; 5Korean Medicine Life Science, University of Science & Technology (UST), Campus of Korean Institute of Oriental Medicine, Daejeon 34054, Republic of Korea

**Keywords:** dry eye disease, USL, Achyranthis radix extract, randomized controlled trial

## Abstract

**Introduction:** Dry eye disease (DED) is a very frequently encountered ocular disease, making it a growing public health burden. However, current treatments for DED present unmet medical needs owing to their side effects or ineffectiveness. Therefore, an effective and safe therapeutic agent to manage DED is needed. **Method and Analysis:** We planned a phase 2, dose-finding, double-blind, randomized placebo-controlled trial to evaluate the efficacy and safety of two different doses of USL (Useul), the extract of Achyranthis Radix, compared with placebo, for DED. USL has been found to protect against DED by inducing tear secretion and improving corneal irregularity via anti-inflammatory effects, which will provide new therapeutic options. One hundred and twenty participants will be enrolled, after assessing the inclusion/exclusion criteria, at Daejeon University Daejeon Korean Medicine Hospital. Enrolled participants will be allocated to standard-dose USL, high-dose USL, or placebo groups in a 1:1:1 ratio and will be required to administer the trial medication twice a day for 12 weeks and visit the clinic five times. For efficacy outcomes, objective endpoints of fluorescein corneal staining score, tear break-up time, Schirmer’s test, and meibomian test and subjective endpoints of Ocular Surface Disease Index, visual analog scale, Standard Patient Evaluation for Eye Dryness-II, and biomarkers will be assessed throughout the trial. Safety will be assessed based on adverse events, vital signs, laboratory tests, visual acuity, and intraocular pressure. **Discussion:** Our study results are expected to provide clinical evidence for the use of DED as an effective and safe agent for DED.

## 1. Introduction

Dry eye disease (DED) is a “multifactorial disease of the tear film and ocular surface that results in discomfort, visual disturbance, and tear film instability, with potential damage to the ocular surface. It is accompanied by an increased osmolarity of the tear film and subacute inflammation of the ocular surface”, as proposed by the Tear Film and Ocular Surface Dry Eye Workshop II [1]. Contrary to the previous definition of DED, it focuses more on its pathogenesis, including tear hyperosmolarity, inflammation, neurosensory abnormalities, and unstable tear films [2], while ocular inflammation is believed to play a central role in the pathogenesis of DED [3,4].

Symptoms of dry eye vary in severity and include ocular discomfort, pain, fatigue, foreign body sensation, burning, and visual disturbances [5]. DED leads to significant impairment of quality of life, including aspects of physical, social, and psychological function, such as depression and anxiety disorders [6,7]. DED affects approximately 10.8–57.4% of the population worldwide [8,9], accounting for one of the most common diseases causing patients to visit ophthalmology clinics [10]. The prevalence of DED increases with age; however, it has also recently increased among young people with modern lifestyles and digital environments, including the intensive use of computers or smartphones, the use of contact lenses, and higher psychological stress [11,12]. Therefore, the impact of DED on public health has become more important, and the economic burden of DED on affected patients is more significant. The annual direct cost per patient is estimated to be USD 783 in the United States, USD 664 in Europe, and USD 530 in Japan [13].

The treatment of DED depends on the disease’s severity and etiology. Artificial tears are widely used as the first-line treatment, with patient education and environmental control. When insufficient, anti-inflammatory therapy (topical corticosteroids or topical Cyclosporine A), oral supplements (essential fatty acids), and gels/ointments are administered. Other options include autologous serum, oral tetracycline, punctual plug/occlusion, contact lenses or goggles, systemic anti-inflammatory medications, and surgery when previous treatment is inadequate [5,14,15]. Artificial tears only relieve symptoms in the short term, without resolving the disease process and causes. Topical cyclosporine A or corticosteroids target inflammatory pathways; however, some ocular side effects, such as temporary redness or burning, make them intolerable for long-term use [16,17]. Autologous serum poses a risk of anemia and blood-borne infection [18] and punctual plugs pose a risk of secondary infection [19]. Over time, sophisticated multiple-action combination formulations have gradually replaced simple water-adding medications. They are molecules or combinations of molecules that can improve the quality and quantity of the tear film components but have limited capabilities to interact with the ocular surface epithelia [20].

Owing to these limitations in DED treatment, an effective and safe treatment is needed. Recently, various options to manage the inflammatory pathways of DED have increased dramatically, and dietary modifications, such as essential fatty acids and complementary medicines, have been proposed as promising candidates for the treatment of DED [15]. In East-Asian countries, USL, the extract of Achyranthis Radix (AR), has been used as an herbal medicine for treating arthritis, osteoporosis, delayed menses, and edema [21,22]. AR is also approved as a functional food by the Ministry of Food and Drug Safety in Korea for its effect on maintaining healthy joints and cartilage. Previous pharmacological studies of AR have revealed its anti-inflammatory efficacy via heme oxygenase-1 induction and nuclear factor-κB suppression in macrophages [23], as well as its antibacterial [24], anti-oxidant [25], and anti-platelet activities [26]. Based on the strong anti-inflammatory properties of AR, we administered AR to patients with DED, anticipating that it may also be effective against ocular surface inflammation. A previous experimental study revealed that AR protected against DED by inducing tear secretion and improving corneal irregularity in a particulate matter-induced DED rat model [27]. The bioactive phytochemicals of AR have been reported to be triterpenoids, polysaccharides, and phytoecdysteroids; we analyzed the major components of USL using high-performance liquid chromatography (HPLC) [28]. In addition, we found that AR does not induce oral toxicity or genotoxicity in in vivo and in vitro testing [29].

In this study, we aimed to explore the efficacy and safety of USL in patients with DED and determine the appropriate dosage of USL required for DED treatment. We planned a randomized, double-blind, placebo-controlled, parallel, phase 2 clinical trial of two different doses of USL compared with placebo. We will measure both objective and subjective outcomes associated with DED and observe anti-inflammatory biomarkers to understand the pathways of USL that show efficacy.

## 2. Method and Analysis

### 2.1. Trial Design

This study is a phase 2, dose-finding, double-blind, randomized placebo-controlled trial to evaluate the efficacy and safety of two different doses of USL, compared with a placebo, for DED treatment. The trial will be conducted at Daejeon University Daejeon Korean Medicine Hospital and is set to start recruitment in January 2025. Participants who sign a written informed consent form after receiving a comprehensive explanation of the trial will be screened, and 120 participants will be enrolled after the eligibility assessment. Eligible subjects will be randomized into three groups in a 1:1:1 allocation ratio: standard-dose USL (1 g/day), high-dose USL (2 g/day), and placebo groups. The subjects would be asked to administer one of the medications twice a day for 12 weeks and visit five times (at screening, baseline, week 6, week 12, and week 14) to evaluate the efficacy and safety outcomes (Figure 1, Table 1). This study follows the guidelines of the Standard Protocol Items: Recommendations for Interventional Trials (SPIRIT) [30] and will be conducted in compliance with ethical principles in the Declaration of Helsinki and Good Clinical Practice guidelines [31].

This study protocol was approved by the Institutional Review Board of Daejeon University Daejeon Korean Medicine Hospital (DJDSKH-23-DR-11), and registered at Clinicaltrials.gov (NCT06016010).

### 2.2. Participants

#### 2.2.1. Inclusion Criteria

Participants who meet the following criteria will be enrolled in this trial: (1) subjects aged over 19 years; (2) subjects with dry eye symptoms (stiffness, foreign body sensation, irritation, bloodshot eyes, itching, blurred vision, pain, etc.) for at least 3 months; (3) subjects who have had both eyes screened, with a corrected visual acuity of 0.2 or more; (4) subjects with a fluorescein corneal staining score (Oxford grading) ≥ 2, tear break-up time (TBUT) ≤ 10, and Schirmer’s test ≤ 10 mm at screening; (5) subjects with an Ocular Surface Disease Index (OSDI) ≥ 23 at screening; and (6) subjects who voluntarily agree and provide written informed consent to participate in the clinical trial.

#### 2.2.2. Exclusion Criteria

The exclusion criteria [32,33] include (1) current or history of ocular disorders (ocular surgery, diseases, and trauma) within 2 months before the study that may affect the study results. These include abnormal eyelid function, such as disorders of the eyelids or eyelashes; ocular allergies or current treatment for allergic diseases of the eye (antihistamine, topical ocular mast cell stabilizer use, etc.); and cicatricial keratoconjunctivitis caused by pterygium, pinguecula, herpetic keratopathy, conjunctival scarring (cicatricial pemphigoid, Steven–Johnson syndrome and alkali damage), neurogenic keratitis, lack of congenital lacrimal gland, corneal transplantation or keratoconus; (2) acute eye inflammation/infection within 3 months before the study; (3) subjects who have undergone eye surgery (including LASIK/LASEK) within 12 months before the study; (4) those who have undergone lacrimal punctal occlusion, cauterization of the punctum, or an intense pulsed light procedure within 3 months before the study; (5) wearing contact lenses within 72 h of screening or planning to wear contact lenses during the study period; (6) intraocular pressure (IOP) > 22 mmHg on more than one side; (7) autoimmune diseases (e.g., Sjögren’s syndrome, Graves’ disease, systemic lupus erythematosus, and rheumatoid arthritis); (8) use of cyclosporine or diquafosol in any form (systemic or topical) within 1 month before the study; (9) current or history of medical conditions (glaucoma, ocular allergy, ocular inflammation/infectious disease, etc.) within 1 month of proposed treatment with topical agents besides artificial tears; (10) those who have taken or are taking steroid drugs or immunosuppressants (azathioprine, tacrolimus, cyclosporine, mycophenolate mofetil, etc.) that may affect immune function within 3 months prior to screening; (11) those who have taken antihypertensive drugs, such as antihistamines and beta-blockers, antidepressants and diuretics that can worsen dry eye; (12) severe abnormal liver function (aspartate transaminase or alanine transaminase values ≥ 2 times the upper limit of normal) and abnormal renal function test results (creatinine values ≥ 2 times the upper limit of normal); (13) genetic disorders, such as glucose–galactose malabsorption, Lapp lactase deficiency, and galactose intolerance; (14) comorbidities that may interrupt the treatment or assessment (cancer and disorders of the liver, kidney, cardiovascular system, psychiatric system, endocrine system, respiratory system, and central nervous system; (15) unregulated hypertension (high blood pressure of 160 mmHg in the condenser or >100 mmHg during relaxation); (16) unregulated diabetes (fasting blood sugar exceeding 180 mg/dL); (17) hepatitis A, hepatitis B, or hepatitis C; (18) history of hypersensitivity reaction to the active ingredients of the investigational medicine; (19) history of excessive alcohol use or drug addiction; (20) pregnancy or lactation; (21) those who did not agree to use medically permitted methods of contraception; (22) those who had participated in other clinical trials within the last 4 weeks; and (23) subjects who are judged as ineligible to participate in this study.

#### 2.2.3. Sample Size

The estimated sample size required for this trial is 120 patients (40 patients per group), based on previous comparable studies [34,35,36]. Although we could not calculate the sample size based on the exact effect size, as our study is the first trial using USL on patients with DED, we calculated the sample size based on previous studies expected to have similar effects. We assumed the difference in the changes in TBUT as 1.3, and the standard deviation (SD) as 1.8 from baseline to post-intervention between the control and standard-dose USL groups. The power to detect differences is assumed to be 0.8, with the two-sided significance level set at 0.05. The ratio of the allocation of subjects between groups is 1:1:1, and the dropout rate is assumed to be 0.2.

#### 2.2.4. Recruitment

Participants will be recruited from the outpatient department of Daejeon University Daejeon Korean Medicine Hospital by posting brochures on the bulletin board and Internet homepage of the clinical trial center. For additional promotions, newspaper leaflets and local advertisements on subways, buses, and apartment notice boards will be considered, if needed.

### 2.3. Randomization and Allocation Concealment

Randomization will be conducted by an independent statistician by using a computer random number generator in SAS^®^ version 9.4 (SAS institute. Inc., Cary, NC, USA). Based on a generated random-number table, the participants will be allocated to the standard-dose USL, high-dose USL, and control groups in a 1:1:1 ratio. The manufacturer, who is responsible for the management of the random allocation sequence, will label the clinical trial medications according to the randomization list and deliver them to the management pharmacist on the trial site. When the investigator requests the trial medication, the management pharmacist will provide the corresponding medication using the participant’s randomization code.

### 2.4. Blinding

All participants and investigators, including outcome assessors, will be blinded throughout the trial. The trial medication was packaged identically with a placebo developed to have the same color, taste, and smell as the USL intervention medicine. The subject identification code will be sealed in an opaque envelope, stored in a locked cabinet, and not disclosed until the end of the trial, except if a serious adverse reaction occurs.

### 2.5. Intervention

The trial medication, USL, and placebo were produced by the National Institute of Korean Medicine Development. USL is an extract of AR, and one tablet (800 mg) of USL contains 500 mg of AR extract. The placebo does not contain the major ingredient of USL but contains starch, lactose, and colorants. Both drugs were manufactured as round and brown tablets. The dosage was determined based on the results of a previous experimental study that reported a lack of toxicity and genotoxicity of USL and the experimental study results, and USL was found to have no toxicity and genotoxicity in a previous study [29]. The standard-dose USL group will be administered one tablet of USL and one tablet of the placebo twice daily for 12 weeks; the high-dose USL group will be administered two tablets of USL twice daily for 12 weeks; and the control group will be administered two tablets of the placebo twice daily for 12 weeks.

Participants will be provided with the trial medication at week 0 and week 6 and asked to return to calculate medication compliance. The overall compliance should be at least 75%, and those with <75% will be excluded from the per-protocol (PP) analysis.

### 2.6. Outcomes

#### 2.6.1. Primary Outcome

The primary outcomes of this trial include both objective and subjective outcome measures recommended by the Ministry of Food and Drug Safety of Korea. For objective outcomes, TBUT was chosen as the primary outcome to assess tear film stability, and for subjective outcomes, OSDI will be assessed to evaluate symptoms and discomfort related to DED.

##### Tear Break-Up Time (TBUT)

TBUT was chosen as the primary outcome for objective signs in our study because TBUT assesses tear film stability, which is one of the essential diagnostic criteria for diagnosing tear film abnormalities and DED [37]. TBUT is also one of the most frequently used diagnostic tests in clinical practice for DED [38].

The procedure follows a standardized methodology. For the TBUT test, subjects will be instructed to sit at the slit-lamp microscope with their chin on the chin rest. Sodium fluorescein will then be applied to both eyes on the inferior temporal bulbar conjunctiva. Each participant will be instructed to blink naturally three times to distribute fluorescein on the ocular surface. Then, the participants will be asked to cease blinking until instructed otherwise. The time between eye opening and the first appearance of a dry spot in the tear film will be measured in seconds using a slit-lamp microscope (BQ-900; Haag-Streit, Bern, Switzerland). The cut-off value of TBUT for the diagnosis of DED is reported to be less than 10 s for an abnormal tear film, between 5 s and 10 s for a marginal tear film, and less than 5 s for severe DED [39]. With reference to this, participants with a TBUT ≤ 10 s will be included in this trial. The TBUT for each participant will be measured at screening, week 6, week 12, and week 14, and we will measure the changes in mean TBUT from screening to week 12 between the intervention and control groups as the primary outcome.

##### Ocular Surface Disease Index (OSDI)

The OSDI is the primary outcome measure of subjective symptoms in this trial. It is the most commonly used questionnaire for patients with DED in clinical trials. The OSDI has been validated to measure the severity of DED, and it comprises psychometric symptoms related to DED [40]. Owing to its strong establishment in DED, a consensus to use the OSDI was reached by a related committee [15].

The OSDI measures the frequency of symptoms, environmental triggers, and vision-related quality of life within 1 week. It comprises 12 questions, each of which is graded on a scale of 0–4, where 0 indicates none of the time, 1 as some of the time, 2 as half of the time, 3 as most of the time, and 4 as all of the time [40]. The total OSDI score is calculated as follows: OSDI = [(sum of scores for all questions answered) × 100]/[(total number of questions answered) × 4]

Thus, the OSDI score ranges from 0 to 100; the higher the score, the worse the quality of life. According to OSDI scores, patients can be categorized as having normal (0 to 12 points), mild (13 to 22 points), moderate (23 to 32 points), or severe (33 to 100 points) ocular surface disease. In our trial, patients with OSDI scores ≥ 23 points will be included, meaning that those who have moderate-to-severe symptoms will all be included. The OSDI score will be measured at screening, week 6, week 12, and week 14, and the changes in OSDI from screening to week 12 will be the primary outcome of this trial.

#### 2.6.2. Secondary Efficacy Outcome

##### Fluorescein Corneal Staining Score (FSC)

Ocular staining is used to diagnose DED, assess its severity, and measure its clinical response to therapy. It is one of the most frequently used outcome measures in clinical trials. In our study, fluorescein staining will be evaluated to measure the severity of ocular surface disease using standardized methods and the Oxford grading system [41]. After fluorescein instillation, both eyes will be examined under a slit-lamp microscope (BQ-900; Haag-Streit, Bern, Switzerland) using standard settings. The same investigator will evaluate the corneal staining score throughout the trial period. The grades will range from 0 to 5: 0, absent (dot count 0); 1, minimal (dot count 10); 2, mild (dot count 32); 3, moderate (dot count 100); 4, marked (dot count > 316); and 5, severe (dot count > 316). The fluorescein corneal staining score is an inclusion criterion in our study, and patients with an Oxford grading ≥ 2 will be included. Fluorescein corneal staining will be conducted at screening, week 6, week 12, and week 14 to observe changes after the administration of trial medications.

##### Schirmer’s Test

Schirmer’s test (without anesthesia) will be performed to measure tear secretion in patients with DED. It will be conducted using a standardized process that involves bending the Schirmer strip at the notch and asking the participant to look up and pull down their lower eyelid. Next, we will hook the bent end of the strip over the center of the lower eyelid, with the eye gently closed for 5 min. The length of wetting from the notch will be measured in millimeters [42]. The Schirmer test without anesthesia is a well-standardized measurement of evaluating stimulated reflex of tear flow and the cut-off values of ≤10 mm/5 min for diagnosing dry eye, as reported in a previous study [39].

##### Meibomian Gland Test

The meibomian gland test will be used to evaluate meibomian gland function in terms of quality and expressibility. In the normal eyelid, the meibum is clear and expressed readily under gentle pressure. However, the meibum becomes opaque and difficult to express in patients with meibomian gland dysfunction [43].

To evaluate the expressibility of meibum, five meibomian glands in the central area of the upper or lower eyelids will be assessed under a slit-lamp microscope after applying firm digital pressure. Patients will be scored from 0 to 3 based on the following criteria: 0, all glands expressible; 1, 3–4 glands expressible; 2, 1–2 glands expressible; and 3, no glands expressible. Meibum quality will be evaluated by observing eight central meibomian glands of the upper and lower eyelids and scored from 0 to 3: 0, clear; 1, cloudy; 2, cloudy and particulate; and 3, opaque toothpaste-like meibum [44]. Both meibum quality and expressibility will be assessed at weeks 0, 6, 12, and 14.

##### Visual Analog Scale (VAS)

The 100 mm visual analog scale (VAS) is a patient-reported outcome that evaluates ocular discomfort, such as ocular pain, itching, dryness, burning, or foreign body sensation. Participants will be scored according to ocular discomfort severity on a 100 mm line, with higher scores representing more severe symptoms. This will be assessed at weeks 0, 6, 12, and 14.

##### Standard Patient Evaluation for Eye Dryness-II (SPEED-II)

The Standard Patient Evaluation for Eye Dryness-II (SPEED-II) is a questionnaire developed to track the progress of dry eye symptoms over time [45]. It is a valid and reliable questionnaire that assesses the severity and frequency of dry eye symptoms, including irritation, dryness, scratchiness, grittiness, burning, soreness, watering, and eye fatigue. The total score ranges from 0 to 28, and dry eye symptoms can be graded as mild (0–4), moderate (5–8), or severe (>8). It monitors three different timeframes: now, within the last 72 h, and within the last 3 months [46]. In this trial, SPEED-II will be measured at weeks 0, 6, 12, and 14.

Compared with the OSDI, SPEED-II includes questions related to the severity of dry eye symptoms, and the symptoms listed in OSDI and SPEED-II are not identical. SPEED-II has a timeframe of 3 months, while OSDI covers 1 week prior to the evaluation [47]. Recently, the use of SPEED-II has increased since the sensitivity and specificity are reported to be 0.90 and 0.80, respectively [48].

#### 2.6.3. Exploratory Efficacy Outcomes

##### Biomarkers (IFN-γ, IL-1β, TNF-α, and MMP-9)

Biomarkers of interferon-gamma (IFN-γ), interleukin-1beta (IL-1β), tumor necrosis factor-α (TNF-α), and matrix metalloproteinase-9 (MMP-9) will be measured to investigate the inflammatory effect of USL on patients with DED. Recent studies have reported that proinflammatory cytokines TNF-α and IL-1β, and anti-inflammatory cytokine IFN-γ play important roles in ameliorating DED. Moreover, IL-1β, IFN-γ, and TNF-α are the most important cytokines used for DED treatment in animal models [49]. MMP-9 is also considered as valuable therapeutic target in DED by playing a vital role in the pathogenesis of corneal disease. Recent clinical and animal studies also introduce MMP-9 as a biomarker of ocular surface inflammation in DED [50]. IFN-γ, IL-1β, and TNF-α will be obtained from each of the subject’s serum and will be analyzed by a designated laboratory, and MMP-9 activity will be measured from tears (InflammaDry; Rapid Pathogen Screening, Inc., Sarasota, FL, USA). It will be evaluated before and after administration of the trial medication (weeks 0 and 12).

#### 2.6.4. Safety Outcomes

Adverse events (AEs), vital signs, laboratory examinations, visual acuity, and IOP will be assessed throughout the trial to evaluate safety. At every visit, AEs, vital signs, visual acuity, and IOP will be observed, and laboratory tests will be performed before and after taking the trial medication. We will record every unexpected symptom or sign during the trial by asking the participants questions and through the investigator’s examination. However, if serious AEs occur during the trial, the patient will be immediately removed from the trial and proper treatment will be provided. Visual acuity and IOP will be examined as recommended by the Ministry of Food and Drug Safety in Korea for clinical trials on ocular diseases. The laboratory tests will include liver function, routine blood, and urine tests.

### 2.7. Statistical Analysis

Statistical analyses of all outcomes will be conducted by an independent statistician using SAS Version 9.4 Analytics Pro software. Continuous variables will be presented as the mean ± SD and categorical variables will be presented as frequencies and percentages. For efficacy analysis, a full analysis set (FAS) based on intention-to-treat (ITT) analysis will be performed as the primary analysis, and PP analysis will be used for secondary analysis. All randomized participants who are evaluated at least once after administration of the trial medication will be included in the FAS, and only those who complete the trial without violating the protocol will be included in the PP set. A safety analysis will be conducted for all participants who are randomized and administered the trial medication at least once.

The primary and secondary efficacy outcomes will be analyzed using the mixed-effects model for repeated measures, with each group and visit set as fixed effects and participants as random effects. In addition, the difference between pre- and post-treatment within groups will be analyzed using Student’s paired *t*-test or Wilcoxon signed-range test, depending on normality. A multiple imputation method will be used for missing values. The efficacy of USL will be accepted at a significance level of 2.5% in both TBUT and OSDI compared with the placebo, whereas other secondary outcomes will be accepted at a 5% significance level.

To analyze safety outcomes, Fisher’s exact test will be used to compare the number of AEs between the groups, and Student’s paired *t*-test or the Wilcoxon signed-rank test will be performed to analyze the differences between pre- and post-treatment laboratory variables, vital signs, visual acuity, and IOP. Demographic characteristics will be analyzed using the Wilcoxon rank-sum test or independent *t*-test for continuous variables and Fisher’s exact test or the chi-square test for categorical variables.

### 2.8. Data Collection, Management, and Monitoring

All clinical trial data will be recorded, processed, and preserved to enable accurate reporting, interpretation, and confirmation in accordance with the Good Clinical Practice guidelines. Investigators will be acquainted with the trial protocol and follow the standard operating procedures of this trial.

Data collected at each visit from the source documents of each participant will be entered into an electronic case report form and managed by a data manager during the trial period. All data will be coded to maintain confidentiality, and only the principal investigator will be able to access the data. Throughout the study, full-source data verification will be conducted by a clinical research associate and the data manager.

Data monitoring will be conducted via regular and occasional visits, based on the monitoring plan. At each monitoring stage, the clinical research associate will check whether the trial is being conducted in compliance with the protocol and related regulations. All data will be collected accurately, and AEs will be properly reported. Any problems encountered during the trial will be resolved by the investigators.

### 2.9. Ethics Approval and Dissemination

All subjects will be provided with all information related to this trial, including the entire trial process, possible benefits and risks of intervention, other therapeutic options for the target disease, and the right to withdraw from the trial. The potential unintended adverse events of intervention, including stomach discomfort, anorexia, nausea, and diarrhea, which are commonly observed in herbal medicines, will also be informed to each subject. After sufficient time and opportunity to ask questions about the trial, participants will be asked to sign a written consent form. Every participant will be immediately notified of any changes or factors that may affect them. After the trial has ended, the study results will be disseminated through peer-reviewed journals and a registry of clinical trials.

## 3. Discussion

Despite the growing burden of DED on public health, current treatments for DED present unmet medical needs owing to their side effects or ineffectiveness. Clinicians and patients with DED seek better treatment therapies to manage inflammation and ocular symptoms; nutritional supplements or complementary medicine have emerged as promising candidates for DED by managing ocular surface inflammation [51]. As inflammation is recognized as a major factor in the pathogenesis of DED, we anticipate that the anti-inflammatory effects of USL would improve the symptoms and signs of DED.

Our trial was designed to evaluate the efficacy and safety of USL for DED compared with a placebo and to determine the appropriate dosage of USL. We plan to conduct a phase 2, double-blind, dose-finding, randomized placebo-controlled trial, and both objective signs and subjective symptoms of DED, as well as cytokines related to inflammation, will be investigated. Objective signs and subjective symptoms are important in the diagnosis and management of DED; however, previous studies have reported a lack of association between them, making DED more complicated. The Food and Drug Administration (FDA) recommends that the efficacy of treatment should be demonstrated by showing significant differences in objective prespecified signs and subjective prespecified symptoms of dry eye. Moreover, the FDA suggests that different endpoints for objective signs (conjunctival staining, corneal staining, Schirmer’s test score, and TBUT) or subjective symptoms should be conducted when designing clinical trials for the treatment of DED. According to the FDA guidelines, we will evaluate TBUT and OSDI as primary outcomes and other objective endpoints, including fluorescein corneal staining score, Schirmer’s test, meibomian gland test, and subjective endpoints of VAS and SPEED-II as secondary outcomes. In addition, to investigate the inflammatory effects of USL, cytokines IL-1β, IFN-γ, and TNF-α will be measured.

This is the first clinical trial to evaluate the efficacy and safety of USL for DED. Along with our results, which will show the effects of USL in inducing tear secretion, improving corneal irregularity, and ameliorating inflammation on the ocular surface, we expect that USL will have effects on DED. USL is safe and has little risk of side effects, as it is already considered a Korean medicine or functional food in Korea. With this experimental and empirical evidence, our study results will provide new evidence for the efficacy and safety of USL on DED in a well-designed trial following the FDA guidelines.

This trial protocol has some limitations. First, as our trial is the first study to evaluate USL in DED, the sample size cannot be calculated based on the exact effect size of USL. Thus, we estimated the effect size based on the results of previous studies in which a similar effect was expected. Second, our trial will be conducted at a single hospital, which can create bias, as a recent epidemiological study of binary outcomes reported that single-center trials showed larger intervention effects than multicenter trials. Recruiting subjects only in Korea may also cause bias in the same context. Third, our exclusion criteria can reduce external validity, though severe restrictions are needed to achieve approval from the Ministry of Food and Drug Safety for clinical trials. Finally, as our trial has a 14-week trial period, maintaining the compliance of the patient is an important issue, and strategies to prevent dropout will be needed. However, we will compensate for this risk by carefully evaluating the methodology and enrolling a sufficient number of participants to demonstrate treatment differences. Further multicenter studies with an accurately calculated number of samples will be conducted to confirm the efficacy and safety of USL.

## Figures and Tables

**Figure 1 healthcare-12-02383-f001:**
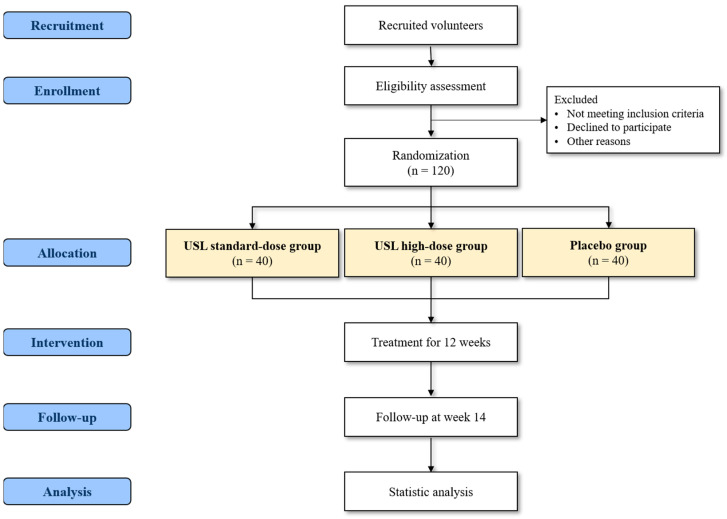
Flowchart of the trial process.

**Table 1 healthcare-12-02383-t001:** Schedule of enrolment, interventions, and assessments.

	Study Period
	Enrolment	Allocation	Post-Allocation	Follow-Up
Visit	1	2	3	4	6
Timepoint	−2 week	0	6 weeks ± 7 days	12 weeks ± 7 days	14 weeks ± 7 days
Enrolment:
Eligibility screen	X				
Informed consent	X				
Demographics	X				
Medical andtreatment history	X				
Physical examination	X				
Vital sign	X	X	X	X	X
Visual acuity, IOP	X	X	X	X	X
Laboratory tests	X			X	
EKG	X				
Allocation		X			
Interventions:
Standard-dose USL		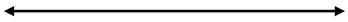	
High-dose USL		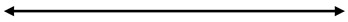	
Placebo		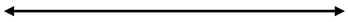	
Assessments:
OSDI	X		X	X	X
TBUT	X		X	X	X
VAS		X	X	X	X
Schirmer’s test		X	X	X	X
Meibomian gland test		X	X	X	X
FCS	X		X	X	X
SPEED-II		X	X	X	X
Biomarkers		X		X	
Adverse events		X	X	X	X
Compliance test			X	X	

“X” indicates procedures occurring at the designated time points. EKG: Elektrokardiogram; IOP: intraocular pressure; OSDI: Ocular Surface Disease Index; TBUT: tear break-up time; VAS: visual analog scale; FCS: fluorescein corneal staining; SPEED-II: Standard Patient Evaluation for Eye Dryness-II; USL: Useul.

## Data Availability

The datasets used or analyzed during the current study will be available from the corresponding author upon reasonable request.

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
