# Peer review of "Efficacy and Safety of Useul for Dry Eye Disease: Protocol for a Randomized, Double-Blind, Placebo-Controlled, Parallel, Phase 2 Clinical Trial"

_healthcare, 2024, doi:10.3390/healthcare12232383_

Round 1

Reviewer 1 Report (Previous Reviewer 2)

Comments and Suggestions for Authors

I would like to sincerely thank the authors for their efforts in updating the manuscript and addressing the concerns raised in the previous review. However, after carefully reviewing the revised version, I still hold the same opinion regarding the main shortcomings of the document.

While I appreciate the necessity and motivation behind registering the study with the ethical committee, and I understand the differences in content between the registration and the manuscript, my concern lies in the substantial similarities between the two. As mentioned in my previous review, this again brings up the question of the potential impact of the current publication. While the study is certainly of great interest, its true value will only be realized once the data collection is completed and the results are fully analyzed. Therefore, I would respectfully suggest that publishing the full results at a later stage would provide a more comprehensive contribution to the scientific community.

Additionally, while the references have been updated, the same issue persists. A significant number of them are over 10 years old, with the most recent one dating to 2021. This remains a critical point, as relying on outdated literature in an ongoing study may lead to findings that do not align with the current state of knowledge, potentially reducing the study’s relevance.

My primary concern also remains regarding the use of these tests as outcome measures, given that they are simultaneously employed as inclusion criteria based on cut-off values. While the inclusion criteria themselves appear appropriate, excluding certain values in the initial analysis could introduce bias into the study sample, potentially compromising the statistical robustness of the findings. As the authors noted in their response, restricting the analysis to patients with moderate to severe DED will limit the degrees of freedom in the statistical tests, as the sample no longer represents the full stochastic variability required for the analysis to confirm or refute the hypothesis.

Regrettably, I must conclude that the revised manuscript has not fully addressed the major concerns outlined in the previous review.

Author Response

While I appreciate the necessity and motivation behind registering the study with the ethical committee, and I understand the differences in content between the registration and the manuscript, my concern lies in the substantial similarities between the two. As mentioned in my previous review, this again brings up the question of the potential impact of the current publication. While the study is certainly of great interest, its true value will only be realized once the data collection is completed and the results are fully analyzed. Therefore, I would respectfully suggest that publishing the full results at a later stage would provide a more comprehensive contribution to the scientific community.

  • Thank you for your thorough opinion. Your concerns in terms of potential impact of protocol paper, may raise issues regarding the publication of ‘study protocol articles’. However, the importance of establishing a protocol at the outset of every clinical trial is unarguable and making protocols publicly available by publishing articles will contribute to improving the transparency of research and will enable researchers and prospective participants to learn about trials that are underway. Protocol availability also enables to understand the scientific rigor of the design and results, as well as to compare what was intended with what was described in the reports of trials in order to assess possible reporting bias. For these reasons, various journals publish ‘study protocols’, as well as Healthcare.
  • The protocol, which is defined as, a document that provides sufficient detail to enable understanding of the background, rationale, objectives, study population, interventions, methods, statistical analyses, ethical considerations, dissemination plans, and administration of the trial; replication of key aspects of trial methods and conduct; and appraisal of the trial’s scientific and ethical rigor from ethics approval to dissemination of results, by “Standard Protocol Items: Recommendations for Interventional Trials (SPIRIT) statement”, can provide sufficient details to understand the general design and major features of the trial, in contrast to the result papers which focus on reporting outcomes.
  • As we recognized your concerns with the similarities between registration and study protocol paper, we followed the SPIRIT to report our protocol. We will be appreciated if you can thoughtfully consider values of protocol for RCT once again. (Li, Tianjing, et al. "Review and publication of protocol submissions to trials–what have we learned in 10 years?." Trials18 (2017): 1-5.)

Additionally, while the references have been updated, the same issue persists. A significant number of them are over 10 years old, with the most recent one dating to 2021. This remains a critical point, as relying on outdated literature in an ongoing study may lead to findings that do not align with the current state of knowledge, potentially reducing the study’s relevance.

  • I updated references to align with the current state.

My primary concern also remains regarding the use of these tests as outcome measures, given that they are simultaneously employed as inclusion criteria based on cut-off values. While the inclusion criteria themselves appear appropriate, excluding certain values in the initial analysis could introduce bias into the study sample, potentially compromising the statistical robustness of the findings. As the authors noted in their response, restricting the analysis to patients with moderate to severe DED will limit the degrees of freedom in the statistical tests, as the sample no longer represents the full stochastic variability required for the analysis to confirm or refute the hypothesis.

  • For developing drugs for dry eye, FDA recommend using both objective sign (measured by corneal staining, conjunctival staining, tear breakup time, Schirmer’s tear test score) and subjective symptom as outcome measure. As these outcomes are also diagnostic tools for dry eye disease, same outcomes are also used in inclusion criteria. For these reasons, many outcome measures are being used to diagnose and measure therapeutic effects. Moreover, drugs for dry eye disease are different according to their severity. Anti-inflammatory therapies are recommended to being used in moderate to severe dry eye disease. These designs can be seen in other clinical trials expect to have similar effects. (A Randomized, Double-Masked, Placebo-Controlled Clinical Trial of Two Forms of Omega-3 Supplements for Treating Dry Eye Disease)

Reviewer 2 Report (New Reviewer)

Comments and Suggestions for Authors

By focusing on a novel anti-inflammatory agent derived from Achyranthis Radix, the authors aim to provide a new therapeutic option with fewer side effects compared to current treatments. The clinical trial protocol is well-structured, however, the background, the description of the methods,  and the discussion should be improved before further consideration for publication.

1. Think about emphasizing in the abstract the trial's distinctive features, including how USL may be able to treat dry eye disease (DED) in a way that is different from existing therapies.

2. The background should be expanded reporting updated references on new therapeutic treatments for dry eye, as the new multiple acting tear substitutes (e.g. https://doi.org/10.3390/biomedicines12091945).

3. The anti-inflammatory properties of USL are discussed in a general manner. The topic might be strengthened by providing further information on the particular inflammatory pathways (such as NF-κB and heme oxygenase-1) that USL targets in ocular inflammation. Consider adding a graphical abstract on USL mechanisms of action.

4. Consider adding the estimation of the effect size of the sample.

5. Explain why you excluded patients with autoimmune diseases (e.g. Sjogren's syndrome, rheumatoid arthritis....) that could benefit from a new DED treatment.

6. Report extensively the documented side effects of USL that have been reported in previous studies, and how they would impact and would be managed in the clinical trial.

7. Expand the section on limitations of the the study in the discussion (generalizability of the findings to populations outside Korea, possibles bias of the study, high rates of exclusion criteria that could reduce external validity, compliance of the patient during a 14-weeks timeline)

Comments on the Quality of English Language

Minor editing is required

Author Response

  1. Think about emphasizing in the abstract the trial's distinctive features, including how USL may be able to treat dry eye disease (DED) in a way that is different from existing therapies.
  • Thank you for recommendations. Providing features of our trial different from current treatment will make authors to easily understand the rationale of our study. I added these rationale in abstract.

  1. The background should be expanded reporting updated references on new therapeutic treatments for dry eye, as the new multiple acting tear substitutes (e.g. https://doi.org/10.3390/biomedicines12091945).
  • I added newly developed therapeutic treatments as multiple tear substitutes and soft preservatives on backgroud. Thank you for your considerable comments.

  1. The anti-inflammatory properties of USL are discussed in a general manner. The topic might be strengthened by providing further information on the particular inflammatory pathways (such as NF-κB and heme oxygenase-1) that USL targets in ocular inflammation. Consider adding a graphical abstract on USL mechanisms of action.
  • I added the USL mechanism in graphic abstract. Thank you for suggesting the way of strengthening our study.

  1. Consider adding the estimation of the effect size of the sample.
  • I added the estimation of the effect size by previous studies expected to have similar effects.

  1. Explain why you excluded patients with autoimmune diseases (e.g. Sjogren's syndrome, rheumatoid arthritis....) that could benefit from a new DED treatment.
  • We excluded autoimmune diseases as these conditions can affect the anti-inflammatory effects in direct or indirect way. As our trial aims to evaluate the efficacy of USL on DED via anti-inflammatory effects, all conditions or drugs that can affect inflammatory effects were excluded for this trial. For other diseases for which anti-inflammatory drugs are expected to be effective, additional trials will needed to be designed.

  1. Report extensively the documented side effects of USL that have been reported in previous studies, and how they would impact and would be managed in the clinical trial.
  • As our trial is the first study the safety and efficacy of USL on human, there are no clinical data about side effects of USL. Also, based on our previous non-clinical data, USL is expected to haves no safety and tolerability problems in humans.

  1. Expand the section on limitations of the the study in the discussion (generalizability of the findings to populations outside Korea, possibles bias of the study, high rates of exclusion criteria that could reduce external validity, compliance of the patient during a 14-weeks timeline)
  • Thank you for providing a broad perspective on our study. Mentioned issues are needed to be considered throughout the trial. We added those issues on our discussion.

Reviewer 3 Report (New Reviewer)

Comments and Suggestions for Authors

Overall, this is a comprehensive Phase 2 clinical trial design.

A Phase 2 trial should have primary readouts that  focus on safety. Exploratory efficacy readout addressing signs and symptoms of dry eye should be evaluated to help power a subsequent Phase 3 registration trial.

As the authors plan on limiting over the counter artificial tears and lubricants, this becomes an explanatory trial. A run in period is recommened (e.g. 2 weeks), in which all subjects receive vehicle, prior to being allocated to either experimental group. 

A note on biomarkers: although MMP9 is a useful diagnostic biomarker, MMP9 levels typically do not respond to treatment. Therefore, MMP9 response (as well as all biomarkers) should be considered only exploratory biomarkers. 

Methods for diagnostic tests could be described in more detail. 

Comments on the Quality of English Language

Minor editing required.

Author Response

As the authors plan on limiting over the counter artificial tears and lubricants, this becomes an explanatory trial. A run in period is recommened (e.g. 2 weeks), in which all subjects receive vehicle, prior to being allocated to either experimental group.

  • Thank you for suggesting us adding ‘run-in period’ for our trial. When we designed our trial, we had also considered to add ‘run-in period’. However, guidelines for developing drugs for dry eye disease did not suggest ‘run-in period’, and other clinical trials for dry eye disease also did not have ‘run-in period’. Instead, we have wash-out period of 1 month for lubricants before trial enrollment and artificial tears are allowed during the trial.(U.S. Department of Health and Human Services Food and Drug Administration Center for Drug Evaluation and Research (CDER). Dry Eye: Developing Drugs for Treatment Guidance for Industry. (2020))

A note on biomarkers: although MMP9 is a useful diagnostic biomarker, MMP9 levels typically do not respond to treatment. Therefore, MMP9 response (as well as all biomarkers) should be considered only exploratory biomarkers.

  • We revised biomarkers to exploratory outcomes. Although we want to evaluate some of anti-inflammatory biomarker, derived from our previous experimental studies or FDA guidelines, biomarkers tend to not respond to treatment as you mentioned. Thank you for your considerable comments.

Methods for diagnostic tests could be described in more detail.  

  • As methods for diagnostic tests are being used for outcome measures, I added detailed procedures of outcomes. 

Round 2

Reviewer 1 Report (Previous Reviewer 2)

Comments and Suggestions for Authors

I want to thank the authors for the detailed response. I appreciate the effort that implies their responses, and the proposal to follow SPIRIT guidelines and the value of protocol publications in enhancing transparency and scientific rigor. However, I remain concerned about the scientific impact of publishing this protocol at this stage, as its full value might only be realized when results are available. Lastly, while I understand the rationale for using both diagnostic and outcome measures, I believe the overlap with inclusion criteria might still introduce bias, potentially affecting the study’s statistical robustness. I want to apologize for my position, as I ultimately maintain my reservations regarding the publication of the protocol at this time. I will leave the final decision and any specific considerations to the journal’s policies and editorial guidelines.

Author Response

[Comments 1]

I want to thank the authors for the detailed response. I appreciate the effort that implies their responses, and the proposal to follow SPIRIT guidelines and the value of protocol publications in enhancing transparency and scientific rigor. However, I remain concerned about the scientific impact of publishing this protocol at this stage, as its full value might only be realized when results are available. Lastly, while I understand the rationale for using both diagnostic and outcome measures, I believe the overlap with inclusion criteria might still introduce bias, potentially affecting the study’s statistical robustness. I want to apologize for my position, as I ultimately maintain my reservations regarding the publication of the protocol at this time. I will leave the final decision and any specific considerations to the journal’s policies and editorial guidelines.

  • Although it was not enough to get your agreement, we thank you for considering our opinion. As we revised the anticipated start date of the trial to January 2025, our trial is not yet in recruitment, which means protocol papers can be submitted and published in this status. Protocol papers are usually acceptable before the trial ended or more rigorously before the trial started. Moreover, our trial design is based on previous clinical trials for dry eye disease and followed FDA recommendation, which usually limit the severity of disease to the level most appropriate for the mechanism of the drug. We hope our explanation will help you understand our intention. Thank you.
  • Below is the diagnose algorithm in TFOS DEWS II Report, which is also included in our inclusion criteria.

Reviewer 2 Report (New Reviewer)

Comments and Suggestions for Authors

Consider changing lines 66-68 (reference 20) with:

"Over time, sophisticated multiple-action combination formulations have gradually replaced simple water-adding medications. They are molecules or combination of molecules that can improve the quality and quantity of the tear film components but have limited capabilities to interact with the ocular surface epithelia". 

This will make the statement more clear and complete.

Comments on the Quality of English Language

Minor English editing required

Author Response

Consider changing lines 66-68 (reference 20) with:"Over time, sophisticated multiple-action combination formulations have gradually replaced simple water-adding medications. They are molecules or combination of molecules that can improve the quality and quantity of the tear film components but have limited capabilities to interact with the ocular surface epithelia". This will make the statement more clear and complete.

  • Thank you for your thorough consideration for our manuscript. Your recommendation makes the statement more clear. We really appreciate for your help in English editing.

This manuscript is a resubmission of an earlier submission. The following is a list of the peer review reports and author responses from that submission.

Round 1

Reviewer 1 Report

Comments and Suggestions for Authors

The manuscript titled "Efficacy and Safety of USL for Dry Eye Disease: Protocol for a Randomized, Double-Blind, Placebo-Controlled, Parallel, Phase 2 Clinical Trial" is well-organized and addresses a clinically relevant issue. The study aims to assess the efficacy and safety of USL in treating dry eye disease (DED), a common condition that significantly affects patients' quality of life. The trial design is comprehensive, but several areas require clarification and minor adjustments:

1.Exclusion Criteria for Intraocular Pressure (IOP) (Lines 134–136):

The exclusion criteria state that patients with IOP over 22 mmHg will be excluded. However, some individuals may have naturally higher IOP due to factors such as a thicker cornea without having glaucoma. I recommend refining this criterion to exclude only those with a confirmed diagnosis of glaucoma, rather than using IOP alone. This will prevent the exclusion of patients who may have high IOP physiologically but are not at risk for glaucoma.

2.Ocular Inflammation Biomarkers (Lines 305–312):

Including inflammation biomarkers such as matrix metalloproteinase-9 (MMP-9) would provide more comprehensive data on the anti-inflammatory effects of USL. This could strengthen the study’s ability to assess the medication’s impact on ocular surface inflammation, which is critical to understanding its efficacy.

3.Control Over Artificial Tear Usage (Lines 314–318):

The use of artificial tears during the study period could confound the results, making it difficult to attribute symptom improvements solely to the trial medication. I recommend limiting or standardizing the use of artificial tears across groups to ensure that any improvement in DED symptoms can be attributed to USL.

4.Simplification of Exploratory Outcome Structure (Lines 313–319):

The section "2.6.3. Exploratory Outcomes" contains only one item and should not be presented as a separate subheading. Integrating this exploratory outcome into the overall outcomes section without a subheading would simplify the structure and improve clarity.

Author Response

Answer’s to Review1

1.Exclusion Criteria for Intraocular Pressure (IOP) (Lines 134–136):

The exclusion criteria state that patients with IOP over 22 mmHg will be excluded. However, some individuals may have naturally higher IOP due to factors such as a thicker cornea without having glaucoma. I recommend refining this criterion to exclude only those with a confirmed diagnosis of glaucoma, rather than using IOP alone. This will prevent the exclusion of patients who may have high IOP physiologically but are not at risk for glaucoma.

-> Thank you for your considerable comment regarding the risks of using intraocular pressure alone. However, we included IOP on our exclusion criteria as high IOP has several factors as surgeon history, glaucoma, diabetes which we think might affect on our clinical outcomes.

2.Ocular Inflammation Biomarkers (Lines 305–312):

Including inflammation biomarkers such as matrix metalloproteinase-9 (MMP-9) would provide more comprehensive data on the anti-inflammatory effects of USL. This could strengthen the study’s ability to assess the medication’s impact on ocular surface inflammation, which is critical to understanding its efficacy.

-> Thank you for recommendation to include inflammatory biomarkers. Reflecting your opinion, we added inflammatory biomarker of MMP-9. Additional explanations are listed in 2.6.2.6 Biomarkers.

3.Control Over Artificial Tear Usage (Lines 314–318):

The use of artificial tears during the study period could confound the results, making it difficult to attribute symptom improvements solely to the trial medication. I recommend limiting or standardizing the use of artificial tears across groups to ensure that any improvement in DED symptoms can be attributed to USL.

-> Regrading the risk of using artificial tears, we decided to limit the use of artificial tears in this trial.

4.Simplification of Exploratory Outcome Structure (Lines 313–319):

The section "2.6.3. Exploratory Outcomes" contains only one item and should not be presented as a separate subheading. Integrating this exploratory outcome into the overall outcomes section without a subheading would simplify the structure and improve clarity.

-> As we limited the use of artificial tears throughout our trial period, this outcome will also be deleted in our trial. Thank you for your thoughtful opinions.

Reviewer 2 Report

Comments and Suggestions for Authors

In the present manuscript, the author outlines a study protocol designed to clinically evaluate the efficacy and safety of two doses of USL, compared with a placebo, in treating dry eye disease over a masked 3-month study period. The manuscript is well-written and effectively structured, offering clear insight into the rationale behind the proposed research. However, as noted by the authors (lines 105-107), the study protocol has already been approved by an ethics committee. It is registered in a clinical trials database, with full details available online at https://clinicaltrials.gov/study/NCT06016010?term=NCT06016010&rank=1. This raises a pertinent question regarding the potential impact of the current publication. While the study is undeniably of great interest, its true value will only be realized upon the study's completion and the analysis of the data collected. Therefore, I would respectfully suggest that the full publication of the study's results, once available, would offer a more comprehensive understanding and contribute greater scientific value to the field.

Moreover, I would kindly recommend a thorough review of the references. The majority are over 10 years old, with the most recent one from 2021. This is particularly important, as using outdated knowledge in an ongoing study may result in findings that are not aligned with the current state of the field, thereby potentially diminishing the study's relevance. In this context, I recommend ensuring consistency in the criteria used for testing and in the rationale behind the study's methodology. For example, while several references are made to the Tear Film and Ocular Surface Dry Eye Workshop II (e.g., lines 36-37), other principles appear to be applied at certain points in the study (e.g., in the inclusion criteria on lines 114-121, which seem more aligned with the Asia Dry Eye Society (PMID: 31425351)). This is not a methodological issue, but providing clarity on the rationale behind these decisions would help readers better understand the study's approach.

Finally, while I acknowledge that the study is funded, I am also aware of the time and effort required for a 120-sample study. Therefore, I would respectfully suggest that the document should indicate whether the recruitment process has begun, in order to justify the publication of this study protocol on its own previous to the results assessment.

Additional Comments:

Line 20: The acronym "USL" should be explained the first time it appears in the text.

Lines 44-45: Consider adding updated data to the text (e.g., PMID: 34545606).

Lines 46-52: The references and data in this section should be updated (e.g., PMID: 34670680, PMID: 37336259).

Lines 53-59: This section lacks sufficient references. It would be helpful to include some, such as PMID: 28736343.

Line 98 and Figure 1: The mention of “standard” and “high” doses of the treatment is noted, but no specific details are provided that would allow for the reproducibility of the protocol. I recommend specifying the composition or concentration used for each group at some point in the document, as this information is essential for understanding the potential impact of future results.

Lines 123-158: There are no references provided to justify the exclusion criteria. These should be added.

Lines 208-211: I’m concerned about the reliability of establishing these tests as outcome measures since they are also used as inclusion criteria based on cut-off values (lines 118-120). While the inclusion criteria seem appropriate, excluding a range of values in the initial analysis could bias the study sample, potentially compromising the statistical reliability of the analysis.

Lines 254-255: Will be the investigator blinded to the results of the other tests? Are they involved in the measurement process?

Lines 431-435: Since the study hasn’t been conducted yet, this section should be written in the future tense.

Author Response

Answer’s to Review2

In the present manuscript, the author outlines a study protocol designed to clinically evaluate the efficacy and safety of two doses of USL, compared with a placebo, in treating dry eye disease over a masked 3-month study period. The manuscript is well-written and effectively structured, offering clear insight into the rationale behind the proposed research. However, as noted by the authors (lines 105-107), the study protocol has already been approved by an ethics committee. It is registered in a clinical trials database, with full details available online at https://clinicaltrials.gov/study/NCT06016010?term=NCT06016010&rank=1. This raises a pertinent question regarding the potential impact of the current publication. While the study is undeniably of great interest, its true value will only be realized upon the study's completion and the analysis of the data collected. Therefore, I would respectfully suggest that the full publication of the study's results, once available, would offer a more comprehensive understanding and contribute greater scientific value to the field.

-> Thank you for your considerable comments with our study. For publication of study protocol, it is mandatory to achieve approval of study by ethics committee and to register at clinical trials database. Our protocol paper includes rationale of our study design along with our previous animal studies, specific trial design and statistical methods which is not presented on registry database. This information will be essential to assess risk of bias on our trial result paper whether we followed the planned design and statistical analysis. Please kindly consider the value of our study protocol. Thank you.

Moreover, I would kindly recommend a thorough review of the references. The majority are over 10 years old, with the most recent one from 2021. This is particularly important, as using outdated knowledge in an ongoing study may result in findings that are not aligned with the current state of the field, thereby potentially diminishing the study's relevance.

-> I updated references throughout the paper to reflect current state of DED.

In this context, I recommend ensuring consistency in the criteria used for testing and in the rationale behind the study's methodology. For example, while several references are made to the Tear Film and Ocular Surface Dry Eye Workshop II (e.g., lines 36-37), other principles appear to be applied at certain points in the study (e.g., in the inclusion criteria on lines 114-121, which seem more aligned with the Asia Dry Eye Society (PMID: 31425351)). This is not a methodological issue, but providing clarity on the rationale behind these decisions would help readers better understand the study's approach.

-> Our study follows the criteria of ‘TFOS DEWS Ⅱ Definition and Classification Report. The Ocular Surface 15, 276-283, 2017’ on diagnosis, classification, and management of DED. Inclusion criteria of this trial, objective outcomes as staining score, TBUT, Schirmer’s test and subjective symptom outcomes of OSDI, is also consistent with the diagnostic tests on ‘TFOS DEWS II Diagnostic Methodology report’(Figure 5). Our trial design (our inclusion, exclusion criteria and efficacy outcomes ) is based on the guideline ‘Dry Eye: Developing Drugs for Treatment. Guidance for Industry (Draft Guidance), FDA Center for Drug Evaluation and Research (CDER), December 2020’, and this guideline base on TFOS DEWS.

Finally, while I acknowledge that the study is funded, I am also aware of the time and effort required for a 120-sample study. Therefore, I would respectfully suggest that the document should indicate whether the recruitment process has begun, in order to justify the publication of this study protocol on its own previous to the results assessment.

-> This study has not started recruitment due to the delay of development of investigational product. We will clearly indicate the status of study. Thank you for remining us.

Additional Comments:

Line 20: The acronym "USL" should be explained the first time it appears in the text.

-> I added the explanation of USL, which is the acronym of Useul (the Korean Medicinal name of AAchyranthis Radix).

Lines 44-45: Consider adding updated data to the text (e.g., PMID: 34545606).

-> Thank you for your updated related articles. I added it as the reference of prevalence of DED.

Lines 46-52: The references and data in this section should be updated (e.g., PMID: 34670680, PMID: 37336259).

-> I replaced the references with recent papers presenting the impact of lifestyles on DED. Reflecting these references, other lifestyle risk factors was added.

Lines 53-59: This section lacks sufficient references. It would be helpful to include some, such as PMID: 28736343.

-> I added several references on the current treatment of DED. Moreover, references for limitations of conventional treatments are replaced with recent review papers (reference 20-23). Thank you for advising us to enhance references.  

Line 98 and Figure 1: The mention of “standard” and “high” doses of the treatment is noted, but no specific details are provided that would allow for the reproducibility of the protocol. I recommend specifying the composition or concentration used for each group at some point in the document, as this information is essential for understanding the potential impact of future results.

-> I added the volume of USL on standard and high doses of USL for authors to understand more clearly. Additional information can be found on line 209-212, which describes that standard group administer a total of 2 tablets (contains 500 mg of AR extract) a day, while high-dose group administer a total of 4 tablets a day.

Lines 123-158: There are no references provided to justify the exclusion criteria. These should be added.

-> I added the references for the exclusion criteria which was base on the guideline of FDA.

Lines 208-211: I’m concerned about the reliability of establishing these tests as outcome measures since they are also used as inclusion criteria based on cut-off values (lines 118-120). While the inclusion criteria seem appropriate, excluding a range of values in the initial analysis could bias the study sample, potentially compromising the statistical reliability of the analysis.

-> The cut-off values in inclusion criteria was added to include only moderate to severe DED patients and exclude mild DED patients, as we expect our intervention will show effects on moderate to severe DED patients via anti-inflammatory activities. However, as your concerns, the initial baseline data will be adjusted on analysis.

Lines 254-255: Will be the investigator blinded to the results of the other tests? Are they involved in the measurement process?

 -> All investigators will be blinded through out the trial and they will only perform the assessment of outcomes without knowing which group participants are belong to. This process is described on line 195-197.

Lines 431-435: Since the study hasn’t been conducted yet, this section should be written in the future tense.

-> Although our trial has not been started yet, these funding supported our whole process in the development of USL to obtain pre-clinical data, design the clinical trial, and approval of ethical committee.
